# 2-[18F]FDG-PET/CT in Cancer of Unknown Primary Tumor—A Retrospective Register-Based Cohort Study

**DOI:** 10.3390/jimaging9090178

**Published:** 2023-08-31

**Authors:** Heidi Rimer, Melina Sofie Jensen, Sara Elisabeth Dahlsgaard-Wallenius, Lise Eckhoff, Peter Thye-Rønn, Charlotte Kristiansen, Malene Grubbe Hildebrandt, Oke Gerke

**Affiliations:** 1Department of Clinical Research, University of Southern Denmark, 5000 Odense, Denmark; 2Department of Nuclear Medicine, University Hospital of Southern Denmark, Lillebælt Hospital, 7100 Vejle, Denmark; 3Department of Oncology, Odense University Hospital, 5000 Odense, Denmark; 4Department of Medicine, Center of Diagnostics, Odense University Hospital, Svendborg Hospital, 5700 Svendborg, Denmark; 5Department of Oncology, University Hospital of Southern Denmark, Lillebælt Hospital, 7100 Vejle, Denmark; 6Department of Nuclear Medicine, Odense University Hospital, 5000 Odense, Denmark; 7Centre for Innovative Medical Technology, Odense University Hospital, 5000 Odense, Denmark

**Keywords:** cancer, 2-[18F]FDG-PET/CT, CT, survival, detection rate, unknown primary site

## Abstract

We investigated the impact of 2-[18F]FDG-PET/CT on detection rate (DR) of the primary tumor and survival in patients with suspected cancer of unknown primary tumor (CUP), comparing it to the conventional diagnostic imaging method, CT. Patients who received a tentative CUP diagnosis at Odense University Hospital from 2014–2017 were included. Patients receiving a 2-[18F]FDG-PET/CT were assigned to the 2-[18F]FDG-PET/CT group and patients receiving a CT only to the CT group. DR was calculated as the proportion of true positive findings of 2-[18F]FDG-PET/CT and CT scans, separately, using biopsy of the primary tumor, autopsy, or clinical decision as reference standard. Survival analyses included Kaplan–Meier estimates and Cox proportional hazards regression adjusted for age, sex, treatment, and propensity score. We included 193 patients. Of these, 159 were in the 2-[18F]FDG-PET/CT group and 34 were in the CT group. DR was 36.5% in the 2-[18F]FDG-PET/CT group and 17.6% in the CT group, respectively (*p* = 0.012). Median survival was 7.4 (95% CI 0.4–98.7) months in the 2-[18F]FDG-PET/CT group and 3.8 (95% CI 0.2–98.1) in the CT group. Survival analysis showed a crude hazard ratio of 0.63 (*p* = 0.024) and an adjusted hazard ratio of 0.68 (*p* = 0.087) for the 2-[18F]FDG-PET/CT group compared with CT. This study found a significantly higher DR of the primary tumor in suspected CUP patients using 2-[18F]FDG-PET/CT compared with patients receiving only CT, with possible immense clinical importance. No significant difference in survival was found, although a possible tendency towards longer survival in the 2-[18F]FDG-PET/CT group was observed.

## 1. Introduction

Cancer of unknown primary tumor (CUP) accounts for about 3–5% of definitive cancer diagnoses [1] and is usually defined as cancer with a biopsy-proven metastasis, but without an identified primary tumor after a standard diagnostic workup [2,3]. Diagnosis of CUP usually includes clinical exams, blood samples, biopsies, computed tomography (CT) scans of the thorax and upper abdomen (new guidelines from 2022 also include the pelvic area [4]), and evaluation by a medical diagnostics team [5,6]. As this disease is usually at an advanced stage when detected, median survival is less than a year when no primary tumor is ever found [1,7,8]. Hence, finding the primary tumor is essential for optimizing and increasing the likelihood of successful treatment in these patients.

Traditional imaging methods, including CT scans, typically have a detection rate (DR) of the primary tumor in suspected CUP patients of 35% or less [9,10,11,12]. However, a recent systematic review and meta-analysis found that early use of 2-deoxy-2-[18F]fluoro-D-glucose positron emission tomography and computed tomography (2-[18F]FDG-PET/CT) resulted in a pooled DR of 41% [6], and other recent studies have found DRs ranging from 40 to 77% [13,14,15,16]. Despite this, 2-[18F]FDG-PET/CT is not currently a mandatory diagnostic procedure for CUP patients in European guidelines [17].

Previous studies comparing the detection rates of 2-[18F]FDG-PET/CT with CT for suspected CUP patients have shown varying results, mostly in favor of 2-[18F]FDG-PET/CT [11,12], but some were not statistically significant or directly contradictory, and the sample sizes in these heterogeneous studies are often small [3,18]. Furthermore, the available evidence investigating the detection rate in 2-[18F]FDG-PET/CT often lacks a comparison with conventional diagnostic imaging. To the best of our knowledge, there are no studies comparing survival between suspected CUP patients who received a 2-[18F]FDG-PET/CT and those who received a CT.

Therefore, the aim of this retrospective register-based cohort study was to investigate the efficiency of 2-[18F]FDG-PET/CT compared with CT among all patients registered with the tentative pathological diagnosis “cancer of unknown primary tumor” at the Odense University Hospital from 2014 to 2017. The specific objectives were to compare the detection rate of the primary tumor and the median survival between the two groups of patients receiving either a 2-[18F]FDG-PET/CT or only a CT, while taking background characteristics (sex, age, location of first hospital visit, previously known cancer, performance status, and comorbidity) into consideration.

## 2. Materials and Methods

The Danish Patient Safety Authority and the Region of Southern Denmark approved this retrospective register-based cohort study. The RECORD statement was applied to ensure the quality of the data reported [19].

### 2.1. Participants

Patients registered with a tentative CUP diagnosis in the pathology system at the Odense University Hospital during the years 2014 to 2017 were eligible. Patients were considered CUP patients if they had a pathologically proven metastasis with an unknown primary tumor at the time of biopsy (these patients are henceforth referred to as CUP patients). The groups of focus in the study were patients who had received a 2-[18F]FDG-PET/CT, regardless of what other scans had been performed, compared with patients who received a CT scan and *not* a 2-[18F]FDG-PET/CT from the same period. 

Patients were excluded if they did not receive any relevant scans (i.e., could neither be placed in the 2-[18F]FDG-PET/CT nor the CT group of the study) or if they had insufficient available data (e.g., lacking essential data such as scan or biopsy details) for inclusion in the main statistical analyses.

### 2.2. Imaging

The 2-[18F]FDG-PET/CT and CT scans were performed as considered relevant to the clinical setting in all patients. The 2-[18F]FDG-PET/CT scans were performed in one of three nuclear medicine departments and the CT scans in one of eight radiologic departments in the Region of Southern Denmark. Due to local practices, the scan methods are only roughly described. 

The 2-[18F]FDG-PET/CT scans compiled with the standard European requirements [20], and the scan fields were typically from base of the skull to mid-thigh. Patients fasted for four to six hours prior to the scan. The tracer 2-[18F]FDG was administered intravenously in a dose of approximately 4 MBq/kg. Patients then rested and rehydrated with water before the scans were performed approximately 60 min (±15 min) after FDG injection. The CT part of the 2-[18F]FDG-PET/CT scans varied between high- and low-dose CT. The scan fields for the stand-alone CT scans were primarily of thorax and abdomen while some also included head and neck. The contrast enhancement scheme for diagnostic CT used in general Optiray 300 (1 mL/kg).

All scans were analyzed by imaging specialists who were assigned to the clinical routine at the day of examination. The 2-[18F]FDG-PET/CT scans performed with low dose CT were analyzed by nuclear physicians only, CT scans were analyzed by radiologists only, and the 2-[18F]FDG-PET/CT performed with high-dose CT were analyzed by both nuclear physicians and radiologists in a collaborative effort.

### 2.3. Data

The data were collected from patient records in the regional journal system (COSMIC). Rana Bahij initiated and MSJ and HR finished the data collection. Any positive findings on a scan that were considered suspect for malignancy were registered as suggestions for the location of the primary tumor. Information on patient characteristics were collected (Table 1 and Table 2). Data included number of comorbidities, sorting these into categories such as “cardiovascular diseases”, “neurological diseases”, “other cancers”, etc. Additional information on final diagnosis, treatment, and death were collected. Appendix A contains details on the biopsy from the metastasis. Follow-up concluded on 25 May 2022.

### 2.4. Endpoints

The primary endpoints of the study were DR and survival. Secondary endpoints were time from first contact with the hospital to a final diagnosis, the number of applied radiological procedures, and the number of cancer patient pathways. A cancer patient pathway is a politically decided, accelerated diagnostic process available for patients with suspected cancer in Denmark.

### 2.5. Definitions

As reference standard for the final diagnosis, we used either (a) biopsy from the primary tumor, (b) findings on autopsy, or (c) clinical decision to either start treatment or stop further diagnostics based on an available plausible diagnosis. In case a definitive final diagnosis was not available (e.g., a specific type of cancer other than CUP was not found), this was marked as either “not obtained” (a clinical decision had been made to stop diagnostics) or “indefinite” (no final decision to stop diagnostics was made). 

The findings of a scan were considered true positive if the scan suggested a location of the primary tumor consistent with the final diagnosis.

### 2.6. Statistical Analysis

Data were described according to their respective data type. Median (min-max) summarized quantitative variables, frequencies and percentages recapped qualitative variables. Unpaired t-tests (or, alternatively, Wilcoxon rank-sum test) and Chi-squared tests (or, alternatively, Fisher’s exact test) served exploratory hypothesis testing in demographic, clinical and treatment variables.

We calculated DR as the proportion of true positive scans in both groups, respectively. Wilson score 95% CI supplemented these estimates, which in turn were combined with the square-and-add method to assess the difference in DR [21]. A respective P-value for the difference between groups was derived from this 95% CI of the difference using the Altman–Bland method [22]. 

Kaplan–Meier estimates and Cox proportional hazards regression were used to compare survival estimates in the 2-[18F]FDG-PET/CT group with the CT group. Statistical analyses on survival were adjusted for sex, age, and potential treatment. To reduce any potential referral bias, propensity scores were integrated in the statistical analysis to adjust for background patient characteristics in the 2-[18F]FDG-PET/CT and the CT group. Included variables in the propensity score derivation were sex, age, location of first hospital visit, number of previously known cancer diagnoses, Eastern Cooperative Oncology Group (ECOG) performance-status score within a month of the first hospital visit (on a scale from 0 to 5, with higher number indicating greater disability), and number of comorbidities. Subgroup analyses on survival, stratified by the presence of a definitive diagnosis, were performed. Furthermore, a subgroup analysis was performed to control the analyses on detection rate for the fact that many patients received both a 2-[18F]FDG-PET/CT and a CT scan. To adjust for the potential bias of patients not receiving a final diagnosis, a subgroup analysis on time to final diagnosis was performed in patients where only a definitive diagnosis was obtained.

Data was analyzed with a statistical level of significance of 5%. All analyses were performed using STATA/BE 17 (StataCorp, College Station, Texas 77845 USA).

## 3. Results

After application of inclusion and exclusion criteria, 193 patients were included in the study. Of these, 159 patients had received a 2-[18F]FDG-PET/CT, while the remaining 34 patients had only received a CT (Figure 1). One-hundred-and-seven of the 159 patients in the 2-[18F]FDG-PET/CT group also received a CT. 

No significant difference was found between the 2-[18F]FDG-PET/CT and CT groups in the participants’ age or sex. There was a slight tendency for patients in the 2-[18F]FDG-PET/CT group to initially have a better performance status and lower number of comorbidities, although the difference was not statistically significant. A comparison of the amount of previous cancer diagnoses between the groups showed slightly fewer in the CT group (Table 1).

Participants in the 2-[18F]FDG-PET/CT group generally experienced more cancer patient pathways and more radiological procedures, whilst also waiting a longer time from their date of first visit to the type of scan specified for their group. There was no significant difference in time from first hospital visit to final diagnosis or time from first hospital visit to first visit at cancer patient pathway. Furthermore, there was no significant difference in the number of organs with metastases found on radiological imaging or in the way that the decision of a final diagnosis was reached (Table 2).

As part of the 2-[18F]FDG-PET/CT scan, 45% of patients in the 2-[18F]FDG-PET/CT group received a high-dose CT scan, while 54% received a low-dose CT scan. Scans with an unknown CT dosage constituted 1% of the group. In the 2-[18F]FDG-PET/CT group, 39% of the patients received a contrast enhanced CT, while 59% of the patients received a CT without contrast and 2% of the scans were of unknown contrast status. 

In the CT group, all scans were high-dose. Of these patients, 88% received a contrast enhanced CT scan, while 9% received a CT scan without contrast and 3% of the scans were of unknown contrast status. In the CT group, all patients received a scan of the abdomen, while 85% also received a scan of the thorax, and 15% of the group additionally received a scan of the neck. When manually comparing the individual scan fields in patients who received both scans, no primary tumors were missed in CT scans that were found in the additional field of view of a 2-[18F]FDG-PET/CT scan.

A higher number of patients in the 2-[18F]FDG-PET/CT group received treatment compared with the CT group, and of these, there was a slight tendency for more of these patients to receive curative treatment although this was not statistically significant. A significantly higher percentage of patients in the 2-[18F]FDG-PET/CT group received radiation therapy. An elaborating analysis on this parameter showed no significant difference in the purpose of this treatment. No other treatment variables revealed any statistically significant differences between the groups (Table 3).

### 3.1. Propensity Score

The propensity score analysis showed a significantly higher statistical likelihood of patients receiving a 2-[18F]FDG-PET/CT, predicted from baseline characteristics in the 2-[18F]FDG-PET/CT group compared to the CT group, as shown in Appendix A. Mean propensity scores were 84% and 74% in the 2-[18F]FDG-PET/CT and CT groups, respectively.

### 3.2. Detection Rate

The DR was significantly higher in the 2-[18F]FDG-PET/CT group (36.5%) compared to the CT group (17.6%). The significance remained in the subgroup analysis which compared the DR of 2-[18F]FDG-PET/CT to CT, in patients who had received both scans (Table 4).

### 3.3. Survival

The mean follow-up time was 1.13 years per patient in the CT group and 1.77 years per patient in the 2-[18F]FDG-PET/CT group. The median survival for patients in the CT group was 3.8 months, which increased to 5.1 months when only patients who received a definitive diagnosis were included. In the 2-[18F]FDG-PET/CT group, median survival was 7.4 months, and 11.4 months for this group of patients with a definitive diagnosis (Table 5, Figure 2). One- and two-year survival for all patients was 42.8% and 25.8% in the 2-[18F]FDG-PET/CT group and 23.5% and 11.8% in the CT group, respectively. However, while the hazard ratio was consistently lower for the 2-[18F]FDG-PET/CT group, the survival advantage in the 2-[18F]FDG-PET/CT group was only significantly improved in the crude model for all patients. In the adjusted models for all patients, significance disappeared when adjusting for age, sex, treatment status, and propensity score (Table 6). In the subgroup analyses, which included patients with and without a definitive diagnosis, no statistically significant difference in survival was observed between these two subgroups, as shown in Appendix A.

## 4. Discussion

### 4.1. Main Findings

This study has found a significantly higher detection rate of the primary tumor in CUP patients who received a 2-[18F]FDG-PET/CT compared to those who received a CT. No significant difference in survival was found. However, a statistically non-significant tendency towards longer survival in the 2-[18F]FDG-PET/CT group was observed.

### 4.2. Strengths and Limitations of the Study

A common problem in non-randomized studies is adjusting for differences between included patients [23]. This is also an apparent challenge in this study, as there has been no way of ensuring that patients who received a 2-[18F]FDG-PET/CT scan and a CT scan were completely comparable. As this is a register-based study, it has the advantage of having a relatively small amount of missing data and no loss to follow-up [23], but the quality of our data depends on the consistency of the medical reports obtained as well as the consistency of the included radiological procedures, where no standardization was possible. As the 2-[18F]FDG-PET/CT scans in our study usually covered a larger field of view than the CT scans, It is possible that the 2-[18F]FDG-PET/CT scans could have detected lesions that were out of range on the CT scans in some patients. However, our comparison of patients receiving both scans showed no difference between the two groups on this matter.

To adjust for any potential selection bias, we calculated a propensity score which described the likelihood of patients receiving a 2-[18F]FDG-PET/CT based on variables present before referral to any scans. As shown in our results, patients receiving a 2-[18F]FDG-PET/CT had a higher propensity score than patients only receiving a CT. Our elaborated analysis on this matter might suggest a slight tendency towards patients of poorer health being more likely to only receive a CT scan, perhaps because CT scans are faster and less of a burden for an already unwell patient. However, data on performance status were unavailable in a relatively large number of patients in both groups. Our results furthermore show a tendency towards a longer time to final diagnosis in the 2-[18F]FDG-PET/CT group, as shown in Appendix A. The reason for this might be that our patients in this group generally experienced a longer time from first hospital visit to receiving a scan. Approximately two thirds of the patients receiving a 2-[18F]FDG-PET/CT also received a CT, which indicates that some patients might have only received a 2-[18F]FDG-PET/CT if a CT scan was inadequate. This may help explain the lower DR for both 2-[18F]FDG-PET/CT and CT scans in patients who received both (Table 4), as well as why patients in the 2-[18F]FDG-PET/CT group generally were included in more cancer patient pathways and had more radiological procedures during their diagnostic process.

As seen in our survival analysis, adjusting the hazard ratio only for propensity score diminished the difference between our 2-[18F]FDG-PET/CT and CT groups, and this difference was greater than when adjusting for both propensity score and other confounding variables. The analysis indicates that age and sex have an independent impact on survival, apart from the impact of the propensity score. Furthermore, as this study finds a significantly higher detection rate in the 2-[18F]FDG-PET/CT group, there is a possibility that more people in this group were able to receive proper treatment due to correct diagnosis of the primary tumor. This is supported by the findings in our survival subgroup analyses stratified by the presence of a definitive diagnosis, which show a longer survival in patients who eventually received a definitive diagnosis, presumably because these patients were able to receive treatment specific to their diagnosis. The impact of treatment on survival proved to be a statistically significant advantage, which was also shown in these analyses (Appendix A). Therefore, considering that this study also found a slight difference in performance status between our two study groups, age, sex, and general health must be considered significant factors in the likelihood of patients in this study (a) receiving proper diagnostic imaging, (b) receiving proper treatment, and (c) surviving longer.

In this study, all CUP patients between 2014 and 2017 were included if they had a relevant scan. This is an advantage when interpreting the results for patients beyond our study, as it represents a great variety of patients and provides a relatively large sample size. Varying definitions of CUP and/or different practices concerning the use of CT and/or 2-[18F]FDG-PET/CT could potentially cause issues with generalization. Due to local tradition at the Odense University Hospital, many patients routinely receive a 2-[18F]FDG-PET/CT scan when having lesions suspected for malignancy, despite this not being the general recommendation in Europe [5,17,24]. This has led our 2-[18F]FDG-PET/CT group to be significantly larger than our CT group which could have resulted in a different distribution of patients, perhaps with different results, if the same type of study were to be conducted elsewhere. The difference in sample sizes could have been eliminated by only including patients who had received both scans and comparing the DR of CT and 2-[18F]FDG-PET/CT within this group; however, this would have made survival analyses comparing the two scans impossible. 

The gold standard in cancer diagnostics is a biopsy of the primary tumor [25]. However, due to the complexity of CUP, a final diagnosis is often obtained through a clinical decision, as is the case with many of the patients included in this study [6,26]. While this might be an accurate depiction of daily clinical practice, this does create an uncertainty regarding the precision of our analyses on detection rate and time to final diagnosis, which has to be taken into consideration when interpreting these results.

In our study, a CUP patient is defined as having at least one pathologically verified metastasis with an unknown primary tumor at the time of biopsy. Other studies, however, define CUP differently and many require that a thorough diagnostic workup has been performed before defining patients as CUP patients [2,11,26,27,28,29]. This has the advantage of filtering out patients with metastases as first sign of malignancy for whom standard diagnostic imaging methods are sufficient for finding the primary tumor. This may provide a more realistic view of the benefit of 2-[18F]FDG-PET/CT for CUP patients, as currently, it is not routinely used early in the diagnostic process [5,17]; however it also makes the scans less directly comparable. 

A few studies did not require a metastasis biopsy to confirm malignancy and included patients with a strong clinical suspicion of cancer as well [2,26,30]. This might improve early diagnosis in patients who would not otherwise be studied, which might lead to a better prognosis [17]. It could also skew the results, however, as these patients may turn out not to have cancer at all. Requiring pathological verification of malignancy, as is the case in this and several other studies, ensures that all included patients have a disease that is relevant to the study [11,27,28,29,31].

### 4.3. Significance and Other Studies

Except for the crude model including all patients, there were no statistically significant differences in survival between the 2-[18F]FDG-PET/CT (median 7.4 months) and the CT groups (median 3.8 months). However, the hazard ratios were consistently lower for the 2-[18F]FDG-PET/CT group, regardless of whether all patients or only patients with or without a definitive diagnosis were included and what adjustments were made for possible confounders (see Table 6 and Appendix A). This might suggest a tendency towards a longer survival in patients who receive a 2-[18F]FDG-PET/CT. Interpretation of this tendency should, however, consider the possible referral bias of the patients in this study. 

Other studies have found the median survival of CUP patients receiving a 2-[18F]FDG-PET/CT to be between 8.4 and 33 months. Unlike our study, these studies counted survival from time of 2-[18F]FDG-PET/CT, and not from the first hospital visit related to the disease [2,26,27,32]. None of these studies compared survival between CUP patients with a CT and CUP patients with a 2-[18F]FDG-PET/CT though, and our literature search indicated a general lack of these types of studies. 

The DR of 36.5% for 2-[18F]FDG-PET/CT was significantly higher than the 17.6% for the CT group. The difference was likewise pronounced when comparing the scans only in patients who had received both scans (Table 4). This suggests that 2-[18F]FDG-PET/CT can detect roughly the same primary tumors as CT and in addition, some primary tumors that CT is unable to detect, which highlights the importance of studies that directly compare these scans. However, the analyses of DR were unadjusted for confounders which might skew the results. This, as well as the difference in sample size between the groups, means that the results, while promising, should be interpreted with caution. 

Other studies have found the DR for 2-[18F]FDG-PET/CT to be between 24% and 75% [2,6,11,12,26,27,28,29,30,31,32,33,34,35,36]. A systematic review from 2017 investigating the DR of 2-[18F]FDG-PET/CT for CUP patients, including 20 studies and a total of 1942 adult patients, found a DR of 41% [6], while another systematic review from 2009 including 11 studies found a pooled DR for 2-[18F]FDG-PET/CT of 37% [31]. These earlier findings support the results found in our study. 

While there is quite a lot of research on the DR of 2-[18F]FDG-PET/CT itself for CUP patients, the same cannot be said for the comparison of 2-[18F]FDG-PET/CT and CT. Two other studies were found that directly compared the performance of these scans in detecting the primary tumors of CUP patients. These two studies reported DRs of 32 and 18% for CT and 28 and 33% for 2-[18F]FDG-PET/CT, respectively [11,12]. Both studies compared scans performed on the same patient. This is similar to what we did in our subgroup analysis, revealing quite comparable results to the study by Gutzeit et al., with higher DRs for 2-[18F]FDG-PET/CT [12]. However, one of the studies was limited to patients with extracervical metastases [11], and the other was quite small with only 45 patients included [12]. Similarly, other studies have attempted to compare CT and 2-[18F]FDG-PET/CT with sample sizes of 36 and 30 in paired designs where both CT and 2-[18F]FDG-PET/CT were applied to each patient [3,18]. With this limited amount of evidence in mind, we believe the contribution of our larger, unpaired study to be of scientific importance.

### 4.4. Perspectives

The fact that our study found a consistently higher DR in 2-[18F]FDG-PET/CT scans compared to CT scans suggests that the diagnostic value of 2-[18F]FDG-PET/CT in CUP patients might be underestimated. Our study shows for the first time a significantly higher DR using 2-[18F]FDG-PET/CT directly compared to CT in a (fairly) large, clinically representative CUP population (using retrospectively collected data). However, according to current Danish and European guidelines, 2-[18F]FDG-PET/CT is not a mandatory procedure for CUP patients [5,17]. This study shows that implementing 2-[18F]FDG-PET/CT in the standard diagnostic workup might be an advantage for CUP patients in general.

According to our research, the amount of evidence regarding survival and the use of 2-[18F]FDG-PET/CT in CUP patients is much smaller [2,26,27,32,34] than that of DR [2,6,11,12,26,27,28,29,30,31,32,33,34,35,36]. While DR might be more easily available for research purposes, survival is of higher significance to the individual patient. Our study has also failed to prove a statistically significant difference in survival between our groups of focus; thus, more studies investigating survival are needed to strengthen the evidence on this subject. Furthermore, a considerable weakness of most studies regarding 2-[18F]FDG-PET/CT scans is data availability. Since randomization is often not possible in this very heterogeneous population, bias may be introduced to the results. 

A prospective (and preferably randomized controlled) trial comparing 2-[18F]FDG-PET/CT with standard diagnostic workup methods would therefore significantly strengthen the amount of evidence on the use of 2-[18F]FDG-PET/CT as a diagnostic tool for CUP patients [37].

## 5. Conclusions

In this retrospective register-based cohort study, we found a significantly higher DR of the primary tumor with 2-[18F]FDG-PET/CT (36.5%) compared toh CT (17.6%) in patients with CUP. No significant difference in survival between the two groups was found. However, a possible tendency towards longer survival in patients who received a 2-[18F]FDG-PET/CT was observed. This study adds to the growing amount of evidence that suggests 2-[18F]FDG-PET/CT as a useful tool when searching for the primary tumor in CUP patients.

## Figures and Tables

**Figure 1 jimaging-09-00178-f001:**
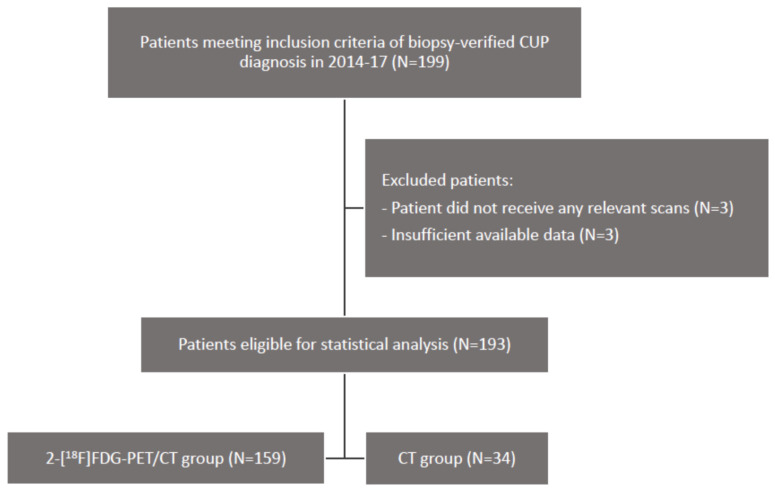
Inclusion and exclusion of CUP patients and division into a 2-[18F]FDG-PET/CT group and a CT group. CUP: Cancer of unknown primary tumor. CT: Computed tomography. 2-[18F]FDG: 2-deoxy-2-[18F]fluoro-D-glucose. PET: Positron emission tomography.

**Figure 2 jimaging-09-00178-f002:**
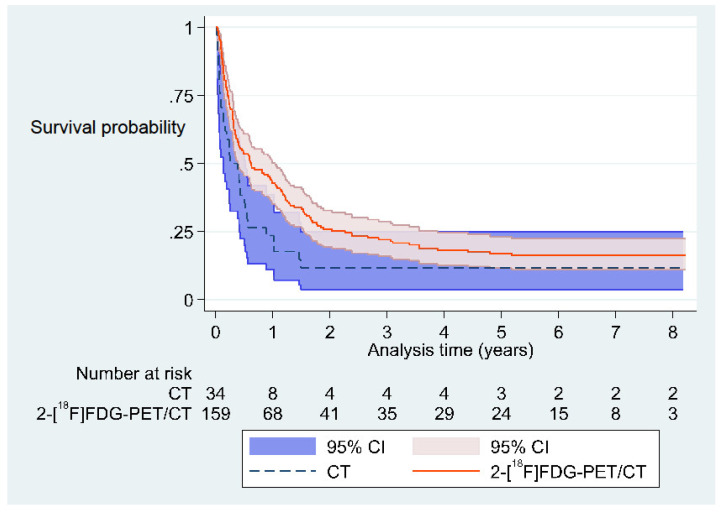
Kaplan–Meier survival estimates in all included CUP patients, comparing patients who received a 2-[18F]FDG-PET/CT with patients who received a CT. CUP: Cancer of unknown primary tumor. CT: Computed tomography. 2-[18F]FDG: 2-deoxy-2-[18F]fluoro-D-glucose. PET: Positron emission tomography.

**Table 1 jimaging-09-00178-t001:** Demographics and baseline characteristics of CUP patients who received a 2-[18F]FDG-PET/CT scan or a CT scan, respectively.

Variable	Subgroup	CT (*N* = 34)	2-[18F]FDG-PET/CT (*N* = 159)	*p*-Value
Sex	Male	21 (61.8%)	80 (50.3%)	0.23 ^#^
	Female	13 (38.2%)	79 (49.7%)	
Median age (min–max)		70.5 (32–95)	68 (29–91)	0.51 ^$^
ECOG Performance status	0	3 (8.8%)	35 (22.0%)	0.12 ^^^
	1	5 (14.7%)	27 (17.0%)	
	2	2 (5.9%)	17 (10.7%)	
	3	7 (20.6%)	13 (8.2%)	
	4	2 (5.9%)	4 (2.5%)	
	Not available	15 (44.1%)	63 (39.6%)	
Median (min–max) number of comorbidities		2 (0–7)	1 (0–7)	0.17 ^+^
Number of previous cancer diagnoses	0	27 (79.4%)	118 (74.2%)	0.046 ^^^
	1	4 (11.8%)	38 (23.9%)	
	2 or >2	3 (8.8%)	3 (1.9%)	

CUP Cancer of unknown primary tumor, CT Computed tomography, 2-[18F]FDG 2-deoxy-2-[18F]fluoro-D-glucose, PET Positron emission tomography. ^#^ Chi-squared test. ^ Fisher’s exact test. ^$^ Unpaired t-test. ^+^ Wilcoxon rank sum test.

**Table 2 jimaging-09-00178-t002:** Clinical characteristics of CUP patients who received a 2-[18F]FDG-PET/CT scan or a CT scan, respectively.

Variable	Subgroup	CT (*N* = 34)	2-[18F]FDG-PET/CT (*N* = 159)	*p*-Value
Number of cancer patient pathways	0	1 (2.9%)	1 (0.6%)	0.001 ^^^
	1	29 (85.3%)	87 (54.7%)	
	2	4 (11.8%)	60 (37.7%)	
	3	0 (0%)	11 (6.9%)	
Median (min-max) number of organs with metastases		2 (1–5)	2 (0–10) ^a^	0.57 ^+^
Median (min-max) number of radiological procedures		2 (1–13)	4 (1–22)	0.004 ^+^
Median (min-max) time from first hospital visit to visit at cancer patient pathway (days) ^b^		2 (0–70)	3.5 (0–130)	0.26 ^$^
Median (min-max) time to scan (days)		0 (0–51)	12 (0–230)	0.001 ^$^
Median (min-max) time to final diagnosis (days) (excluding patients without definitive diagnosis) CT (*N* = 19) 2-[18F]FDG-PET/CT (*N* = 107)		21 (7–250)	33 (5–484)	0.37 ^$^
Basis for final diagnosis (excluding patients marked ‘indefinite’) CT (*N* = 29) 2-[18F]FDG-PET/CT (*N* = 152)	Clinical decision	22 (75.9%)	123 (80.9%)	0.67 ^^^
	Biopsy of the primary tumor	7 (24.1%)	26 (17.1%)	
	Autopsy	0 (0%)	3 (2.0%)	

CUP Cancer of unknown primary tumor, CT Computed tomography, 2-[18F]FDG 2-deoxy-2-[18F]fluoro-D-glucose, PET Positron emission tomography. ^a^ In three patients what was initially thought to be a metastasis was subsequently determined to be the most likely site of the primary cancer. ^b^ Patients with no cancer patient pathways were excluded in both the CT group (*N* = 1) and the 2-[18F]FDG-PET/CT group (*N* = 1). ^^^ Fisher’s exact test. ^$^ Unpaired t-test. ^+^ Wilcoxon rank sum test.

**Table 3 jimaging-09-00178-t003:** Comparison of treatment variables in CUP patients who received a 2-[18F]FDG-PET/CT scan or a CT scan, respectively (using a Chi-squared test).

Treatment Variable	CT (*N* = 33) ^a^	2-[18F]FDG-PET/CT (*N* = 158) ^a^	*p*-Value
Number of patients who received treatment	17 (51.5%)	109 (69.0%)	0.054
Treatment purpose ^b,c^	Curative: 3 (17.6%) Palliative: 14 (82.4%)	Curative: 35 (32.1%) Palliative: 73 (67.0%)	0.22
Chemotherapy ^b^	13 (76.5%)	75 (68.8%)	0.52
Radiation therapy ^b^	3 (17.6%)	61 (56.0%)	0.003
	Curative: 0 Palliative: 3 (100%)	Curative: 18 (29.5%) Palliative: 43 (70.5%)	0.27
Immune therapy ^b^	0	13 (11.9%)	0.13
Surgical treatment ^b^	1 (5.9%)	13 (11.9%)	0.46
Other treatments ^b^	4 (23.5%)	15 (13.8%)	0.30

CUP Cancer of unknown primary tumor, CT Computed tomography, 2-[18F]FDG 2-deoxy-2-[18F]fluoro-D-glucose, PET Positron emission tomography. ^a^ Patients with unknown treatment status were excluded in both the CT group (*N* = 1) and the 2-[18F]FDG-PET/CT group (*N* = 1). ^b^ Only patients who received treatment included. ^c^ Patients with unknown treatment purpose excluded (*N* = 1 in 2-[18F]FDG-PET/CT group).

**Table 4 jimaging-09-00178-t004:** Comparison of detection rates of the primary tumor in CUP patients who received a 2-[18F]FDG-PET/CT scan or a CT scan, respectively.

Groups	CT		2-[18F]FDG-PET/CT		Difference		*p*-Value
	DR	95% CI	DR	95% CI	DR	95% CI	
Patients with *either* a CT (*N* = 34) *or* a 2-[18F]FDG-PET/CT (*N* = 159) available	17.6%	8.3–33.5% ^¤^	36.5%	29.4–44.2% ^¤^	18.8%	1.5–30.9% ^§^	0.012 ^#^
Patients with *both* a CT *and* a 2-[18F]FDG-PET/CT available (*N* = 107)	14.0%	8.7–21.8% ^¤^	32.7%	24.6–42.1% ^¤^	18.7%	7.4–29.5% ^§^	<0.001 ^#^

CUP Cancer of unknown primary tumor, CT Computed tomography, 2-[18F]FDG 2-deoxy-2-[18F]fluoro-D-glucose, PET Positron emission tomography, DR Detection rate. ^¤^ Wilson score 95% CI. ^§^ Square-and-add method. ^#^ Altman–Bland method to obtain the *p*-value from a 95% CI.

**Table 5 jimaging-09-00178-t005:** Overview on survival amongst CUP patients who received a 2-[18F]FDG-PET/CT scan or a CT scan, respectively.

Group	CT	2-[18F]FDG-PET/CT	Total
Number of patients	34	159	193
- Number of patients with definitive diagnosis	19 (55.9%)	107 (67.3%)	126 (65.3%)
- Number of patients without definitive diagnosis	15 (44.1%)	52 (32.7%)	67 (34.7%)
Total follow-up time (years)	38.5	282.1	320.6
- Patients with definitive diagnosis	28.8	223.3	252.1
- Patients without definitive diagnosis	9.7	58.8	68.5
Median (min-max) survival (months)	3.8 (0.2–98.1)	7.4 (0.4–98.7)	6.2 (0.2–98.7)
- Patients with definitive diagnosis	5.1 (0.6–98.1)	11.4 (0.4–98.7)	9.8 (0.4–98.7)
- Patients without definitive diagnosis	2.8 (0.2–59.3)	4.0 (0.4–96.4)	3.8 (0.2–96.4)

CUP Cancer of unknown primary tumor, CT Computed tomography, 2-[18F]FDG 2-deoxy-2-[18F]fluoro-D-glucose, PET Positron emission tomography.

**Table 6 jimaging-09-00178-t006:** Cox-regression regarding survival in all included CUP patients, presenting the hazard ratio of patients who received a 2-[18F]FDG-PET/CT with patients who received a CT as baseline.

Variable	Crude Univariable Model			Multivariable Model 1: Adjusted for Propensity Score			Multivariable Model 2: Adjusted for Age, Sex, Treatment and Propensity Score		
	HR	95% CI	*p*-Value	HR	95% CI	*p*-Value	HR	95% CI	*p*-Value
PET/CT	0.63	0.42–0.94	0.024	0.82	0.54–1.26	0.37	0.68	0.44–1.06	0.087
Age							1.01	1.00–1.03	0.079
Sex (male)							1.24	0.89–1.72	0.20
Treatment (yes)							0.33	0.23–0.48	<0.001
Treatment (unknown)							0.11	0.01–0.78	0.028
Propensity score				0.11	0.04–0.38	<0.001	0.36	0.10–1.38	0.14

CUP Cancer of unknown primary tumor, CT Computed tomography, 2-[18F]FDG 2-deoxy-2-[18F]fluoro-D-glucose, PET Positron emission tomography, HR Hazard ratio.

## Data Availability

The data of this study cannot be shared due to legal restrictions.

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
