# Peer review of "2-[18F]FDG-PET/CT in Cancer of Unknown Primary Tumor—A Retrospective Register-Based Cohort Study"

_2313-433X, 2023, doi:10.3390/jimaging9090178_

Round 1
Reviewer 1 Report
The paper aims to compare 2-[18F]FDG-PET/CT and CT only in CUP patients primarily based on survival and detection rate. Different from the majority of studies utilizing CT scans from PET/CT, the dataset for this paper comprises patient data from CT only scans.
My main concern for this study is the control of factors between the two groups being studied. However, it is worth noting that this limitation is well summarized in the discussion section. Furthermore, the authors have taken steps to adjust the model and minimize the influence of these unrelated factors in the survival analysis.
Overall, the paper has clear organization and fluent language, demonstrating its well-written nature. The topic aligns well with the scope of the journal, and I would recommend its consideration with some minor modifications.
Specific comments (MA = major, MI = minor, OP = optional)
1. [MI] P1, L44. Could you please incorporate the comparison of the median survival values when the primary tumor is identified?
2. [MI] P2, L59-60. From my understanding, one unique aspect of your dataset is the inclusion of CT-only cases, rather than relying solely on CT scans from PET/CT. May I inquire whether you could briefly discuss the advantages of this approach in the paper? Particularly, highlighting the trade-off of having less data for CT-only cases compared to using CT scans from PET/CT.
3. [MI] P2, L80. Could you please emphasize that the first group, who received a 2-[18F]FDG-PET/CT, might also undergo a CT scan? I initially assumed the first group only received a 2-[18F]FDG-PET/CT, but I realized this when reading the later draft.
4. [MI] Table 2. Could you please enhance the format or adjust the text in order to improve the readability of the table, especially the "Variable" column? It is currently challenging to comprehend the variable name and its corresponding value due to some variables spanning across multiple rows.
5. [MI] P4, L126. The use of "N=33" in Table 3 is not explicitly clear whether it refers to the entire CT group or specifically to patients without cancer patient pathways in the CT group. Based on the context, it seems to indicate the former, but to ensure clarity, please specify whether "N=33" refers to the entire CT group. In contrast, "N=1" in Table 3 (P6, L217) seems to pertain to the latter (patients with unknown treatment in the CT group).
6. [MI] P5, L173-180. I believe it would be more appropriate to move this part to the 2.4 Data subsection.
7. [MI] P6, L197-198. Could you please add it to the table as well?
8. [MI] P6, L199. Would the difference in CT dose between the two groups (CT only: all high dose, PET/CT: 45% high dose, 54% low dose) have an impact on the results and analysis of the paper?
9. [MI] Table 3. The format of Table 2 presents a similar issue, as it makes it hard to align variable names with values. For instance, it is unclear what distinguishes the two rows: 13 (76.5%) and 3 (17.6%).
10. [MI] P6, L217. Please clarify "N=1" and make the format the same as Table 2 (P4, L126).
11. [MI] Table 4. The group names are confusing. The first group appears to refer to patients who underwent CT or PET/CT, while the second group pertains to patients who underwent both. Please try to make it more straightforward. Another question is why the CT DR and PET/CT DR in the second row (patients who underwent both CT and PET/CT) are lower than in the first row (patients who underwent at least one of them).
12. [MI] Table 5. Why is there no p-value provided in this table?
13. [MI] Figure 2. Please add a label to the y-axis. Additionally, I would like to inquire about the reason behind the CT curve reaching flat faster than the PET/CT curve. Is this attributed solely to the smaller sample size in the CT group?
14. [MI] P8, L278-L279. What is the sample size for each group? Additionally, how is "no difference" quantified or defined in this sentence?
Author Response
Comments and Suggestions for Authors
The paper aims to compare 2-[18F]FDG-PET/CT and CT only in CUP patients primarily based on survival and detection rate. Different from the majority of studies utilizing CT scans from PET/CT, the dataset for this paper comprises patient data from CT only scans.
My main concern for this study is the control of factors between the two groups being studied. However, it is worth noting that this limitation is well summarized in the discussion section. Furthermore, the authors have taken steps to adjust the model and minimize the influence of these unrelated factors in the survival analysis.
Overall, the paper has clear organization and fluent language, demonstrating its well-written nature. The topic aligns well with the scope of the journal, and I would recommend its consideration with some minor modifications.
Specific comments (MA = major, MI = minor, OP = optional)
- [MI] P1, L44. Could you please incorporate the comparison of the median survival values when the primary tumor is identified?
RESPONSE: Unfortunately, the way that median survival has been reported in the various sources for this study is so different from source to source that it is not possible to report median survival when no primary tumor is ever found with median survival when a primary tumor is found without adding an extensive disclaimer. The authors estimated that this was not necessary in the introduction section which is also the reason for the relatively broad description “less than a year” that is reported here.
- [MI] P2, L59-60. From my understanding, one unique aspect of your dataset is the inclusion of CT-only cases, rather than relying solely on CT scans from PET/CT. May I inquire whether you could briefly discuss the advantages of this approach in the paper? Particularly, highlighting the trade-off of having less data for CT-only cases compared to using CT scans from PET/CT.
RESPONSE: A paragraph has been added to the discussion section noting the importance of having separate study groups if a survival analysis is to be conducted. Regarding our study, a separate CT interpretation was unfortunately not always available in the 2-[18F]FDG-PET/CT scans, thus making it impossible to do a direct comparison within each patient.
- [MI] P2, L80. Could you please emphasize that the first group, who received a 2-[18F]FDG-PET/CT, might also undergo a CT scan? I initially assumed the first group only received a 2-[18F]FDG-PET/CT, but I realized this when reading the later draft.
RESPONSE: A clarification has been added to the beginning of the methods section.
- [MI] Table 2. Could you please enhance the format or adjust the text in order to improve the readability of the table, especially the "Variable" column? It is currently challenging to comprehend the variable name and its corresponding value due to some variables spanning across multiple rows.
RESPONSE: Formatting has been changed, so that additional spacing now separates the variables.
- [MI] P4, L126. The use of "N=33" in Table 3 is not explicitly clear whether it refers to the entire CT group or specifically to patients without cancer patient pathways in the CT group. Based on the context, it seems to indicate the former, but to ensure clarity, please specify whether "N=33" refers to the entire CT group. In contrast, "N=1" in Table 3 (P6, L217) seems to pertain to the latter (patients with unknown treatment in the CT group).
RESPONSE: The numbers have been changed so that the explanations are consistent throughout the manuscript.
- [MI] P5, L173-180. I believe it would be more appropriate to move this part to the 2.4 Data subsection.
RESPONSE: The authors are aware that there might be certain discrepancies in the usual practice when reporting these data, but according to the RECORD statement, which this study complies with, data regarding the numbers of included patients in each stage of the study must be reported in the results section of the paper.
- [MI] P6, L197-198. Could you please add it to the table as well?
RESPONSE: The authors understand and appreciate the suggestion, but as the descriptive tables in the manuscript are characteristics of the included patients and do not address data regarding the scan procedures, it is in our opinion more consistent to leave this in a separate paragraph.
- [MI] P6, L199. Would the difference in CT dose between the two groups (CT only: all high dose, PET/CT: 45% high dose, 54% low dose) have an impact on the results and analysis of the paper?
RESPONSE: The authors recognize the potential confounding factor of the variations in CT dose. However, it is most likely that this will have no practical impact on the interpretation of the scans. It is possible to make a subgroup analysis investigating this matter, but the authors think that this will deviate too much from the original research question.
- [MI] Table 3. The format of Table 2 presents a similar issue, as it makes it hard to align variable names with values. For instance, it is unclear what distinguishes the two rows: 13 (76.5%) and 3 (17.6%).
RESPONSE: Formatting has been changed, so that additional spacing now separates the variables.
- [MI] P6, L217. Please clarify "N=1" and make the format the same as Table 2 (P4, L126).
RESPONSE: The format has been changed and clarification attempted.
- [MI] Table 4. The group names are confusing. The first group appears to refer to patients who underwent CT or PET/CT, while the second group pertains to patients who underwent both. Please try to make it more straightforward. Another question is why the CT DR and PET/CT DR in the second row (patients who underwent both CT and PET/CT) are lower than in the first row (patients who underwent at least one of them).
RESPONSE: The group names have been changed to clarify the difference of the groups. While we do not know if the detection rates differ significantly between the separated and combined groups, a potential significant difference may be due to the fact that patients receiving multiple scans might have tumors that are more difficult to detect as mentioned in the discussion. A reference to the results found in Table 4 has been added to the paragraph in the discussion section concerning this matter.
- [MI] Table 5. Why is there no p-value provided in this table?
RESPONSE: Table 5 serves a descriptive purpose without any inferential intent. Admittedly, there is no consensus in the literature as to add or exclude p-values in descriptive tables; we prefer to exclude p-values here.
- [MI] Figure 2. Please add a label to the y-axis. Additionally, I would like to inquire about the reason behind the CT curve reaching flat faster than the PET/CT curve. Is this attributed solely to the smaller sample size in the CT group?
RESPONSE: We added a label to the y-axis as requested. Moreover, it is possible that the small sample size in the CT group has contributed to the lack of significance in the difference of survival estimates between the groups. However, the tendency of the shorter survival in the CT group is more likely to be attributed to an actual difference between the groups in question or to confounding factors such as performance status (see also Discussion section).
- [MI] P8, L278-L279. What is the sample size for each group? Additionally, how is "no difference" quantified or defined in this sentence?
RESPONSE: We deleted the respective sentence.
Reviewer 2 Report
The enclosed manuscript, drafted by Rimer et al., intends to address the diagnostic value of FDG compared to ordinary CT in the cases of cancer of unknown primary tumor (CUP). Despite the comprehensive statistical analysis being provided, I don't find a common interest to the readers regarding this study's unspecific intention and rationale. There is a fundamental issue hidden behind the experimental design: the nature of the prescription of CT or PET/CT to the patients. Considering the nature of CUP, it is unpredictable in terms of the organ of origin as well as a typical pattern of pathogenic symptoms for diagnostic imaging methods. For instance, how likely that PET/CT is prescribed to patients with suspected cancer lesions, whereas CT is prescribed to patients with less possibility of cancer burdens? Without clear explanations, I don't find the scientific soundness of this research in all aspects, especially when some former studies have revealed the differences (ref #11-12).
Also, some obvious discrepancies in imaging protocols, machines, and post-processing methods were not carefully addressed. It is understandable that the certain law regulation in Denmark might hinder this retrospective study; however, it could be better to at least narrow down the differences between enrolled cases. Otherwise, the results can't be convincing, regardless of the significance.
Unless the authors can carefully address the aforementioned concerns raised, this research can't trigger any interest at the moment.
The language is fine for reading and understanding.
Author Response
Comments and Suggestions for Authors
The enclosed manuscript, drafted by Rimer et al., intends to address the diagnostic value of FDG compared to ordinary CT in the cases of cancer of unknown primary tumor (CUP). Despite the comprehensive statistical analysis being provided, I don't find a common interest to the readers regarding this study's unspecific intention and rationale. There is a fundamental issue hidden behind the experimental design: the nature of the prescription of CT or PET/CT to the patients. Considering the nature of CUP, it is unpredictable in terms of the organ of origin as well as a typical pattern of pathogenic symptoms for diagnostic imaging methods. For instance, how likely that PET/CT is prescribed to patients with suspected cancer lesions, whereas CT is prescribed to patients with less possibility of cancer burdens? Without clear explanations, I don't find the scientific soundness of this research in all aspects, especially when some former studies have revealed the differences (ref #11-12).
Also, some obvious discrepancies in imaging protocols, machines, and post-processing methods were not carefully addressed. It is understandable that the certain law regulation in Denmark might hinder this retrospective study; however, it could be better to at least narrow down the differences between enrolled cases. Otherwise, the results can't be convincing, regardless of the significance.
Unless the authors can carefully address the aforementioned concerns raised, this research can't trigger any interest at the moment.
RESPONSE: The authors agree with the concerns stated regarding the heterogeneous nature of CUP patients. However, as this is a general characteristic of CUP, it would be difficult to eliminate this problem with any study design, apart from a randomized controlled trial that is sufficiently powered to secure balanced groups with respect to demographic and clinical characteristics. We believe though that both necessary sample sizes and ethical concerns hinder the conduct of a randomized controlled trial in CUP. We have discussed this and have taken measures to describe and adjust for bias and confounders (i.e. performance status and difference in protocol) to the extent that was possible in our study.
Reviewer 3 Report
The study conducted by the authors involved a comparison between the use of 2-[18F]FDG-PET/CT and CT alone in terms of their effectiveness in detecting the primary tumour and influencing the overall survival of patients with unknown primary tumour (CUP). The findings of the study indicated that the utilisation of 2-[18F]FDG-PET/CT resulted in a significantly higher detection rate of the primary tumour, and there was also a tendency towards longer survival in the group that underwent 2-[18F]FDG-PET/CT. I would like to provide my comments:
1. I did not comprehend the underlying justification for the comparison of overall survival rates among patients who underwent varying imaging scanning protocols. It is evident that either of these two protocols may potentially exert a direct impact on survival. The authors of the study have raised the possibility that variations in treatment selection, among other factors, following the imaging protocol could potentially impact overall survival. Therefore, it is arguable that comparing survival outcomes in this particular study may not be essential.
2. Certain tables within this manuscript pose difficulties in terms of legibility, particularly table 1 and table 4. I suggest adding additional bottom borders within Table 1 in order to enhance its clarity. In regard to table 4, the column headings are denoted as CT, 2-[18F]FDG-PET/CT, and the titles of the first two rows also correspond to CT, 2-[18F]FDG-PET/CT, which appeared incongruous to my understanding.
3. Section 2.1 was labelled as "study design"; however, this section does not provide any information pertaining to the study design.
4. The utilisation of an asterisk (*) as a superscript marker in tables, particularly in the P value column, can lead to confusion, as it typically denotes a significant difference.
none.
Author Response
Comments and Suggestions for Authors
The study conducted by the authors involved a comparison between the use of 2-[18F]FDG-PET/CT and CT alone in terms of their effectiveness in detecting the primary tumour and influencing the overall survival of patients with unknown primary tumour (CUP). The findings of the study indicated that the utilisation of 2-[18F]FDG-PET/CT resulted in a significantly higher detection rate of the primary tumour, and there was also a tendency towards longer survival in the group that underwent 2-[18F]FDG-PET/CT. I would like to provide my comments:
- I did not comprehend the underlying justification for the comparison of overall survival rates among patients who underwent varying imaging scanning protocols. It is evident that either of these two protocols may potentially exert a direct impact on survival. The authors of the study have raised the possibility that variations in treatment selection, among other factors, following the imaging protocol could potentially impact overall survival. Therefore, it is arguable that comparing survival outcomes in this particular study may not be essential.
RESPONSE: The authors agree that there might be many factors potentially impacting survival, as was discussed in the study as well. However, because the disease progression of CUP patients varies tremendously from patient to patient, selecting homogenous groups eligible for statistical analysis is unfortunately impossible. This study has utilized the retrospective data available to the best of our abilities while recognizing that any data on survival calculated on this complicated subject will be at risk of bias and confounding due to the nature of CUP. A sufficiently powered randomized controlled trial, on the contrary, that secures balanced groups with respect to demographic and clinical characteristics is challenging (if not impossible) due to respective necessary sample sizes and ethical concerns. To this end, we hope to contribute valuable information on survival differences between the groups nonetheless.
- Certain tables within this manuscript pose difficulties in terms of legibility, particularly table 1 and table 4. I suggest adding additional bottom borders within Table 1 in order to enhance its clarity. In regard to table 4, the column headings are denoted as CT, 2-[18F]FDG-PET/CT, and the titles of the first two rows also correspond to CT, 2-[18F]FDG-PET/CT, which appeared incongruous to my understanding.
RESPONSE: Formatting has been changed, so that additional spacing now separates the variables. The group names in table 4 have been changed to clarify the difference of the groups.T
- Section 2.1 was labelled as "study design"; however, this section does not provide any information pertaining to the study design.
RESPONSE: The label has been removed from the section.
- The utilisation of an asterisk (*) as a superscript marker in tables, particularly in the P value column, can lead to confusion, as it typically denotes a significant difference.
RESPONSE: The asterisk has been changed to another symbol.
Round 2
Reviewer 2 Report
Although the authors have made some more additional descriptions regarding my previous comments, the novelty and outstanding findings are still unclear. Something bothering the reading is the conclusion where it states "a significant higher DR of the primary tumor ... in patients with CUP." I am not sure what the "primary" tumor in CUP means, and how it was assessed. Also, it is not novel when using PET/CT for early cancer diagnosis, not limited to CUP. The authors should find a better way to emphasize the scientific importance of this study, or at least, spend a paragraph in introduction or discussion to highlight the unmet needs in CUP diagnosis.
The English writing is clear and readable.
Author Response
Thank you for your thorough review. We have now emphasized throughout the manuscript that patients contributed to either the CT or FDG PET/CT arm, thereby enabling the estimation of a potential survival difference in the first place, whereas previous literature findings are usually based on paired designs in which both modalities are applied to each patient. The definition of when we considered a patient to be a CUP patient is given in section 2.1 (Participants), and Table 2 describes the basis for the final diagnosis. We have extended the discussion sections 4.3 (Significance and Other Studies) and 4.4 (Perspectives) as requested to emphasize the scientific contribution of our study to what is known in the literature.
Reviewer 3 Report
My comments have been addressed and I am happy with the current form of the manuscript.
Author Response
We are glad to learn that we addressed your comments sufficiently and exhaustively. Thank you.